# Hypoxia-Inducible Factor Inhibitors Derived from Marine Products Suppress a Murine Model of Neovascular Retinopathy

**DOI:** 10.3390/nu12041055

**Published:** 2020-04-10

**Authors:** Chiho Shoda, Yukihiro Miwa, Kazumi Nimura, Kazutoshi Okamoto, Satoru Yamagami, Kazuo Tsubota, Toshihide Kurihara

**Affiliations:** 1Laboratory of Photobiology, Keio University School of Medicine, Tokyo 160-8582, Japan; syouda.chiho@nihon-u.ac.jp (C.S.); yukihiro226@gmail.com (Y.M.); 2Department of Ophthalmology, Keio University School of Medicine, Tokyo 160-8582, Japan; 3Department of Ophthalmology, Nihon University School of Medicine, Tokyo 173-8610, Japan; yamagami.satoru@nihon-u.ac.jp; 4Shizuoka Prefectural Research Institute of Fishery and Ocean, Shizuoka 425-0032, Japan; kazumi1_nimura@pref.shizuoka.lg.jp (K.N.); kazutoshi1_okamoto@pref.shizuoka.lg.jp (K.O.); 5Tsubota Laboratory, Inc., Tokyo 160-8582, Japan

**Keywords:** HIF, marine products, retinal neovascularization, oxygen-induced retinopathy

## Abstract

Neovascular retinal degenerative diseases are the leading causes of blindness in developed countries. Anti-vascular endothelial growth factor (VEGF) therapy is commonly used to treat these diseases currently. However, recent reports indicate that long term suppression of VEGF in the eye is associated with chorioretinal atrophy. Therefore, a physiological amount of VEGF is required for retinal homeostasis. Hypoxia-inducible factor (HIF) is a transcriptional factor upstream of VEGF. We previously reported that HIF regulated pathological angiogenesis in the retina of murine models of oxygen-induced retinopathy and laser-induced choroidal neovascularization. Most of the known HIF inhibitors are anti-cancer agents which may have systemic adverse effects in for clinical use; thus, there is a need for safer and less invasive HIF inhibitors. In this study, we screened marine products, especially fish ingredients, and found that six species of fish had HIF inhibitory effects. Among them, administration of *Decapterus tabl* ingredients significantly suppressed retinal neovascular tufts by inhibiting HIF expression in a murine oxygen-induced retinopathy model. These results indicate that particular fish ingredients can act as anti-angiogenic agents in retinal neovascularization diseases.

## 1. Introduction

Pathological retinal angiogenesis is a major pathology of various eye diseases such as diabetic retinopathy (DR), which is one of the most common complications of diabetes, and retinopathy of prematurity (ROP), which is a complication in low birth-weight infants [1,2,3]. These diseases are leading causes of blindness worldwide [4,5]. Neovascular retinopathy has two pathological phases; the first phase is vessel loss leading to tissue ischemia and hypoxia, followed by upregulation of angiogenic factors including vascular endothelial growth factor (VEGF), which stimulates pathological neovascularization in the second phase [2,6,7,8]. Abnormal neovascularization can result in vision loss caused by edema, hemorrhage, retinal fibrosis, scarring, and retinal detachment. Anti-VEGF therapy has been established and is now commonly employed to treat this pathological angiogenesis [9]. However, local or systemic adverse events such as chorioretinal atrophy and renal injury have recently been reported as resulting from potent long-term pharmacological VEGF antagonism [10,11,12]. This is supported by the biological evidence that VEGF is required to maintain physiological vascular homeostasis [13]. Therefore, there is a need to establish a novel therapy for suppressing pathological amount of VEGF without affecting the physiological amount.

We have focused on hypoxia-inducible factors (HIFs), which are transcriptional factors that regulate various genes to adapt to cellular hypoxia [14]. Under normoxic conditions, the subunit of HIFs (HIF-as) is immediately hydroxylated by prolyl hydroxylase (PHD) and ubiquitinated by von Hippel–Lindau protein (VHL) to be degraded in a proteasome-dependent manner [15]. Under hypoxic conditions, the activity of PHD decreases, resulting in HIF-as stabilization, then HIF-as translocate to the nucleus to bind to the hypoxia response element (HRE) in target genes such as VEGF, B-cell lymphoma 2 (BCL2) interacting protein 3 (BNIP3), and phosphoinositide-dependent kinase 1 (PDK1) [16]. We previously reported that pharmacological inhibition of HIFs suppressed retinal neovascularization in murine models of oxygen-induced retinopathy (OIR), known as a retinal neovascular degeneration model [17], and laser-induced choroidal neovascularization (CNV), known as an exudative age-related macular degeneration model [18]. On the other hand, most of the existing HIF inhibitors are anticancer agents [19] which may have systemic side effects in clinical use. Thus, we also need to develop safer and less invasive HIF inhibitors.

Recently, we examined 238 natural products to discover novel HIF inhibitors, and reported that halofuginone extracted from hydrangea has a retinal neuroprotective effect in a murine ischemia–reperfusion model [20]. In the study, fish ingredients such as fish protein from *Spratelloides gracilis* and bio-active shark cartilage powder were also found to suppress HIF activity. There have been some reports about the usefulness of fish ingredients to prevent various diseases. Omega-3 (w-3) polyunsaturated fatty acids (PUFA) from fish oil known as eicosapentaenoic acid (EPA) or docosahexaenoic acid (DHA) are reported to suppress cardiovascular events [21], and these fatty acids also decreased the risk of sight loss in diabetic retinopathy in clinical research [22]. On the other hand, there has been no report about the effect of water-soluble components of fish on ophthalmic diseases.

In this study, we explored water-soluble ingredients from 68 marine species showing HIF inhibitory effects. We also evaluated the therapeutic effects of HIF inhibitors derived from fish on pathological angiogenesis in a murine retinal neovascular degeneration model.

## 2. Materials and Methods

### 2.1. Marine Product Preparation

The material extraction was performed by referring to the protocol previously described [23]. The materials used in this study are shown in Table A1 and A2. Almost all marine product samples were obtained in Shizuoka prefecture, Japan, except *S. gracilis*, which was obtained in Kagoshima prefecture, Japan. All samples were stocked in a freezer (−40 ℃) until extraction. Samples were excised from the dorsal part, fillet, the headless body, and other parts described in Table A1 and Table A2. Each muscle was minced using a knife, and two grams of mince were homogenized with 20 mL cold ultrapure water (generated from Direct-Q3UV, Merck KGaA, Darmstadt, Germany) using a blender (NS-50 and NS-10, Microtec co. ltd, Chiba, Japan) for 2 min. The homogenate was incubated for 30 min in boiling water. After cooling on ice, the homogenate was centrifuged at 1650× *g* for 20 min at 4 ℃. The precipitate was homogenized with 10 mL ultrapure water using a glass rod and centrifuged as described above. These supernatants were filtered using a paper filter (Advantec No. 5A, Toyo Roshi, ltd, Tokyo, Japan) under reduced-pressure conditions, and then a small volume of the oil layer was removed from the filtrate with 10 mL *n*-Hexane. The filtrate was frozen and then dried in a vacuum.

### 2.2. Luciferase Assay for Fish Screening

The luciferase assay was performed as previously described [20]. Human retinal pigment epithelium cell line ARPE19 and murine cone photoreceptor cell line 661W were transfected with a HIF-luciferase reporter gene construct (Cignal Lenti HIF Reporter, Qiagen, Venlo, The Netherlands). The HIF-luciferase construct encodes firefly luciferase gene under the control of a hypoxia response element which binds HIFs. These cells were also co-transfected with a cytomegalovirus (CMV)-renilla luciferase construct as an internal control. These cells were seeded at 1.0 × 10^4^ cells/well/70 mL (ARPE19) or 0.8 × 10^4^ cells/well/70 mL (661W) in an HTS Transwell^®^-96 Receiver Plate, White, tissue-culture (TC)-Treated, Sterile (Corning, NY, USA). At 24 h after seeding, CoCl_2_ (200 mM, cobalt (II) chloride hexahydrate, Wako, Japan) or dimethyloxalylglycine (DMOG) (1 mM, N-(2-Methoxy-2-oxoacetyl) glycine methyl ester, Merck, Darmstadt, Germany) was administered to the cells in order to induce normoxic HIF activation. To evaluate the suppressive effect of fish ingredients against HIF activation, fish ingredients from 69 species were administered at the same time when CoCl_2_ or DMOG was added. After incubation for 24 h at 37 ℃ in a 5% CO_2_ incubator, the luminescence was measured using the Dual-Luciferase^®^ Reporter Assay System (Promega, Madison, WI, USA). Additionally, 1mM of topotecan (Cayman Chemical, Ann Arbor, MI, USA) or doxorubicin (Tokyo Chemical Industry Co., Ltd., Tokyo, Japan) was used as a positive control as known HIF inhibitors.

### 2.3. Real-Time PCR

Total RNA was isolated from the ARPE19 cell line using TRI reagent^®^ (MRC Global, Cincinnati, OH, USA) and an Econospin column for RNA (GeneDesign, Osaka, Japan). The columns were washed with Buffer RPE and RWT (Qiagen, Hilden, Netherlands). RT-PCR was performed using ReverTra Ace^®^ qPCR RT Master Mix with gDNA remover (TOYOBO, Osaka, Japan). Real-time PCR was performed using THUNDERBIRD^®^ SYBR^®^ qPCR Mix (TOYOBO, Osaka, Japan) with the StepOnePlus Real-Time PCR system (Applied Biosystems, Waltham, MS, USA). The primer sequences were as follows: HIF-1a forward TTCACCTGAGCCTAATAGTCC, HIF-1a reverse CAAGTCTAAATCTGTGTCCTG, HIF-2a forward CGGAGGTGTTCTATGAGCTGG, HIF-2a reverse AGCTTGTGTGTTCGCAGGAA, VEGF forward TCTACCTCCACCATGCCAAGT, VEGF reverse GATGATTCTGCCCTCCTCCTT, APO2 forward TCATTAGCCACTGAGTGTTGTTT, APO2 reverse CTCGAATACGATGACTCGGTG, EPO forward CCCTGCCAGACTTCTACGG, EPO reverse GGAGGCCGAGAATATCACGAC, BNIP3 forward GGACAGAGTAGTTCCAGAGGCAGTTC, BNIP3 reverse GGTGTGCATTTCCACATCAAACAT, PDK1 forward ACAAGGAGAGCTTCGGGGTGGATC, PDK1 reverse CCACGTCGCAGTTTGGATTTATGC, GLUT1 forward CGGGCCAAGAGTGTGCTAAA, GLUT1 reverse TGACGATACCGGAGCCAATG, GAPDH forward TCCCTGAGCTGAACGGGAAG, GAPDH reverse GGAGGAGTGGGTGTCGCTGT.

### 2.4. Western Blotting

661W or ARPE19 cells were collected in radioimmunoprecipitation (RIPA) Buffer (Thermo Fisher Scientific, Waltham MA, USA) containing protease inhibitor cocktail (Roche Diagnostics, Basel, Switzerland). Each sample was fractionated by 10% SDS-PAGE and transferred to a polyvinylidene fluoride (PVDF) membrane, then blocked with 5% nonfat dry milk for 1 h at room temperature. The membranes were incubated with primary antibodies: rabbit monoclonal antibodies against HIF-1a (1:1000, Cell Signaling Technology, Danvers, MA, USA), rabbit polyclonal antibodies against HIF-2a (1:1000, NOVUS Biologicals, Centennial, CO, USA) over two nights or mouse monoclonal antibodies against b-actin (1:10,0000, Sigma-Aldrich, St Louis, MO, USA) overnight at 4℃. After washing with tris buffered saline and Tween 20 (TBS-T), the membranes were incubated with horseradish peroxidase-conjugated secondary antibody goat anti-rabbit IgG (1:5000, GE Healthcare, Princeton, NJ, USA) for HIF-1a and HIF-2a or with sheep anti-mouse IgG (1:10,000, GE Healthcare, USA) for b-actin for 1 h at room temperature. The signals were detected using EzWestLumi plus (Atto, Tokyo, Japan). Protein bands were visualized via chemiluminescence (ImageQuant LAS 4000 mini, GE Healthcare, Chicago, IL, USA).

### 2.5. Animals

All procedures related to animal experiments were performed in accordance with the National Institutes of Health (NIH) guidelines for work with laboratory animals, the Association for Research in Vision and Ophthalmology (ARVO) statement for the Use of Animals in Ophthalmic and Vision Research and Animal Research: Reporting In Vivo Experiments (ARRVIVE) guidelines, and were approved by the Institutional Animal Care and Use Committee of Keio University. C57BL/6J mice were obtained from CLEA Japan (Tokyo, Japan).

### 2.6. Oxygen-Induced Retinopathy Model and Administration of Fish Gradients

The OIR model was produced as previously described [17,24]. Postnatal day 8 (P8) mice were exposed to 85% O_2_ for 72 h in an oxygen supply chamber with their nursing mothers. After oxygen exposure, mice were placed back in room air until P17. Pups received oral administration of *S. gracilis* (1.2 g/kg/day), *D. tabl* (3 g/kg/day), or ultrapure water as vehicle once a day from P12 to P16. At P17, the mice were sacrificed, and the eyes were enucleated. The eyes were fixed for 15 min in 4% PFA (paraformaldehyde) solution. Retinal wholemounts were post-fixed in 4% PFA for 1 h. After washing, the tissues were stained with isolectin GS-IB4 from *Griffonia simplicifolia* conjugated to Alexa Fluor 594 (Invitrogen, Carlsbad, CA, USA) at 4 ℃ for 3 days. After encapsulation, retinal vessels were observed with a fluorescence microscope (BZ-9000, KEYENCE, Osaka, Japan). We measured the number of pixels in neovascular tufts and vaso-obliteration using the lasso tool and the magic wand tool of Photoshop (Adobe, San Jose, CA, USA), respectively [25].

### 2.7. Statistical Analysis

We used a two-tailed Student’s *t*-test for comparison of two groups and ANOVA-Turkey for the comparison of three or more groups, respectively. We considered p < 0.05 as being statistically significant. All results in this paper are expressed as the mean + standard deviation.

## 3. Results

### 3.1. In Vitro Screening for Hypoxia-Inducible Factor (HIF) Inhibitors from Marine Products

We prepared 68 types of marine products for screening. All the marine products were water soluble and were dissolved in ultrapure water for use in the experiment. As the first screening, the murine retinal cone cell line (661W) was used to evaluate suppression of HIF activity via HIF luciferase dual assay as previously reported [20]. Cobalt chloride (CoCl_2_) was used to inhibit prolyl hydroxylase (PHD), resulting in an induction of HIF activity, and the suppressive effects of these marine products were then evaluated (Table A1 and Table A2). In the first screening, 27 species showed HIF suppressive effects when compared to vehicle administration under CoCl_2_ exposure (Table A1). These marine products were further examined at the second screening (Table 1 and Table A3). Since it was possible that sequestration of cobaltous ions by chelation with fish ingredients was the cause of HIF inhibition [26], we used dimethyloxalylglycine (DMOG) as another PHD inhibitor in the second screening. Through the second screening, four species of fishes, *Selar crumenophthalmus*, *Spratelloides gracilis*, *Seriola dumerili*, and *Decapterus macarellus* showed significant HIF inhibitory effects when compared with vehicle administration under DMOG stimulation (Figure 1A). The human retinal pigment epithelium cell line (ARPE19) was also used to evaluate the effects of these fish and genealogically related species of fish, *Decapterus muroadsi* and *Decapterus tabl* (Figure 1B). As a result, the screened four and related two species of fish ingredients significantly inhibited HIF activity induced by DMOG. We also evaluated the HIF inhibitory effects of *S. gracillis* at various concentration using the murine embryo fibroblast cell line (NIH-3T3) (Figure A1). *S. gracilis* inhibited HIF activity induced by 1% oxygen in a dose-dependent manner. Further, *S. gracilis* showed a significant HIF inhibitory effect only at a concentration of 1 mg/mL.

### 3.2. Screened Fish Ingredients Inhibit HIF and HIF Target Genes In Vitro

In order to determine how the fish ingredients affect HIF and HIF target genes, ARPE19 cells incubated in 1% oxygen conditions and four species of fish ingredients were added simultaneously. In the ARPE19 cells, the gene expression level of *hif-1a*, decreased by hypoxia (possibly due to a negative feedback [27]), and *hif-2a* was suppressed by fish ingredients (Figure 2A). Expression of HIF target genes such as *vegf*, *epo*, and *pdk1* was upregulated under 1% O_2_ conditions and was significantly suppressed by fish ingredient administration (Figure 2B). Western blotting showed that the protein levels of HIF-1α and HIF-2α in ARPE19 cells, increased by CoCl_2_ (Figure 3A–C), were suppressed by fish ingredient administration. The protein level of HIF-1α in ARPE19 cells, increased by 1% O_2_ (Figure 3D,E), or in 661W cells, increased by CoCl_2_ (Figure 3F,G) or 1% oxygen (Figure 3H,I), was also suppressed by fish ingredient administration. These results indicated that the screened fish ingredients had inhibitory effects on the stabilized HIF expression in pseudo and real hypoxic conditions.

### 3.3. Fish Ingredients Suppressed Neovascularization in a Murine Oxygen-Induced Retinopathy (OIR) Model

To assess the effect of the fish ingredients on retinal neovascularization, we orally administrated them to OIR mice and analyzed neovascular tufts and vaso-obliteration via retinal wholemount staining. Firstly, *S. gracilis* was assessed as a candidate screened in the previous study [20]. Vehicle (*n* = 6) or *S. gracilis* ingredient (1.2 g/kg/day, *n* = 5) was orally administered according to the schedule shown in Figure 4A. There was no significant difference in body weight between the two groups throughout the administration period (Figure 4B). Administration of *S. gracilis* showed little change in neovascular tufts compared to the control (*p* = 0.2) (Figure 4C,D), probably due to technical limitations to increasing the dose of the active ingredient in the crude sample. Thus, in the following experiment, we analyzed *D. tabl*, which was screened in the current study and showed a more potent inhibitory effect on HIF1-α. Vehicle (*n* = 4) or *D. tabl* ingredient (3 g/kg/day, *n* = 4) was orally administered according to the schedule shown in Figure 5A. There was no significant difference in body weight between the two groups throughout the administration period (Figure 5B). Administration of *D.tabl* significantly (*p* < 0.05) suppressed neovascular tufts compared to the control, while no significant difference was observed in vaso-obliteration (Figure 5C,D).

## 4. Discussion

In this study, among marine products from 68 species, we found fish ingredients from four species which had HIF inhibitory effects by luciferase assay (Table 1 and Table A1, Table A2 and Table A3, Figure 1A). Additionally, two species of fish genealogically related to the four species also had HIF inhibitory effects (Figure 1B). These fish ingredients suppressed gene expression of *hif-2a*, followed by suppression of their downstream angiogenic factors and others in vitro (Figure 2A,2B), and the fish ingredients also inhibited HIF-1α and HIF-2α protein expression induced by CoCl_2_ or hypoxia (Figure 3). The activity of HIF-as can be inhibited at the levels of transcription, translation, translocation to the nucleus, and DNA binding [16]. In this study, we found that the fish ingredients suppressed mRNA expression of *hif-2a*. In contrast, the gene expression of *hif-1a* had already been decreased by hypoxia possibly due to a negative feedback [27], and the suppression of *hif-1a* mRNA expression by the fish ingredients could not been seen. At this point, we could confirm that the fish ingredients inhibited HIF-1α and HIF-2α protein expression; however, further mechanism should be investigated in the future studies.

The in vivo experiment revealed that administration of *D. tabl* in an OIR model had a significant suppressive effect on pathological retinal neovascularization (Figure 5). On the other hand, *S. gracilis* showed little change in neovascular tufts and none in vaso-obliteration (Figure 4). In this study, oral administration of ingredients from *S. gracilis* and *D. tabl* was performed at the highest concentration and volume as much as possible according to the procedure previously described [28]. These fish ingredients were crude, and the dosage of *S. gracilis* may not have been sufficient for this model. VEGF is the primary factor driving the formation of neovascular tufts in the OIR model. Although these fish products showed *vegf* suppressive effects in vitro concomitantly with HIF inhibition as well as topotecan which showed a significant suppression of upregulated *vegf* in OIR retinas [17], the changes of VEGF expression level in vivo need to be investigated in the future studies.

The six species of fish are classified into two families: *Spratelloides gracilis* belongs to the Herring family, and *Selar crumenophthalmus*, *Seriola dumerili*, *Decapterus macarellus*, *Decapterus muroadsi*, and *Decapterus tabl* belong to the Carandiae family. Since the fish have similar properties in the same families, it is possible that any characteristic compounds contained in these fishes inhibit HIF activity. There are some reports regarding the disease-preventive effects of w-3 PUFAs derived from fish oil by inhibiting HIF-1a and its downstream pathway. For instance, in a murine model of lung carcinoma, DHA suppressed expression of the HIF-1a/VEGF axis and decreased tumor size with cisplatin treatment [29]. Another report suggested that DHA and EPA attenuated HIF-dependent inflammation and reduced neuronal damage in stroke [30]. In this study, the fish ingredients were incubated in boiled water, then oil in the fish was removed by hexane extraction. We examined HIF activity of each ingredient from some of fishes with or without degreasing with *n*-Hexane using the HIF-reporter luciferase assay, and confirmed that the ingredients containing oil component showed no change in HIF inhibitory effect compared with the oil-free ingredients (Figure A2). Therefore, it is inferred that oil components excluded by this extraction methods have no HIF inhibitory effect, whereas oil-free and water-soluble components contain the biologically active substances. Additionally, the active ingredients contained in these fishes are considered to be small molecules such as dipeptides, amino acids, nucleic acids, and minerals. Further purification of these fish ingredients will be needed. It is also suggested that these water-soluble components do not affect the postnatal growth in OIR mice, as indicated that no change in body weight was observed with administration of either fish ingredients (Figure 4B and Figure 5B). Further studies are needed in order to assess the other physiological responds to these ingredients.

Although anti-VEGF drugs are the main pharmacological approach for macular edema and neovascularization in DR and retinal vein occlusion, and for exudative age-related macular degeneration, long-term VEGF antagonism may induce photoreceptor and Retinal pigment epithelium (RPE) cell atrophy [10,11]. Furthermore, VEGF gene deletion in RPE was shown to induce photoreceptor and choroidal degeneration [18,31]. On the other hand, HIF gene deletion in the retina in adult mice showed no phenotypic change [18], while HIF-as gene deletion in RPE suppressed laser-induced CNV in mice [18]. These data suggest that anti-VEGF drugs may suppress the physiological amount of VEGF required to maintain normal vasculatures and metabolism of cells in the retina and choroid, and that inhibition of HIFs prevents only pathological angiogenesis. Moreover, frequent intravitreal injection of anti-VEGF agents is invasive and of high cost for patients. Therefore, fish ingredients and their active components are readily accepted because of their safety and accessibility for oral intake, and they can be used as a preventive medicine or supplement for proliferative retinopathy.

## 5. Conclusions

We found six types of fish ingredients as novel HIF inhibitors. *D. tabl* had a suppressive effect against pathological retinal neovascularization in a murine OIR model. In conclusion, our results indicate that administration of these fish ingredients may be a possible approach to cure retinal angiogenic diseases by inhibiting HIFs in the retina.

## 6. Patents

The current data includes patents applied for Keio University for a therapeutic or prophylactic agent for ischemic disease, glaucoma, optic nerve disease, retinal degenerative disease, angiogenic retinal disease, cancer, neurodegenerative or autoimmune disease, and a hypoxia inducing factor inhibitor (application no. PCT/JP2017/040884) and by Keio University and Shizuoka Prefectural Research Institute of Fishery for control of hypoxic response by components from marine products (application no. PCT/JP2019/68141, PCT/JP2019/145435).

## Figures and Tables

**Figure 1 nutrients-12-01055-f001:**
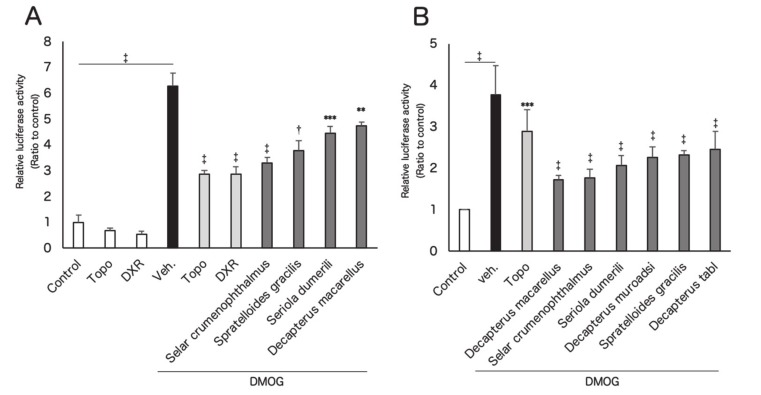
Fish ingredients show inhibitory effects on hypoxia-inducible factor (HIF) activation in vitro. HIF-reporter luciferase assay was performed using the murine retinal cone cell line (661W) (**A**) and the human retinal pigment epithelium cell line (ARPE19) (**B**) cell lines (*n* = 3). Topotecan, doxorubicin, and fish ingredients were administrated in dimethyloxalylglycine (DMOG)-induced culture conditions. Note that six species of fish ingredients significantly inhibited HIF activity induced by DMOG. ** *p* < 0.01, *** *p* < 0.001, † *p* < 0.0001, ‡ *p* < 0.00001 compared with DMOG-Veh. Error bars indicate mean plus SD. Veh., Vehicle; Topo, topotecan; DXR, doxorubicin.

**Figure 2 nutrients-12-01055-f002:**
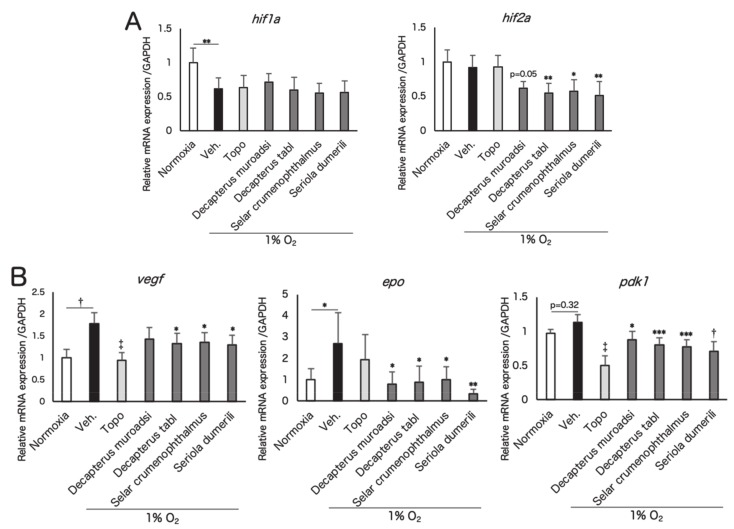
Fish ingredients suppress HIFs and their target genes associated with angiogenesis in vitro. Real-time PCR was performed for *hif-1a* and *hif-2a* (**A**) and their target genes, including *vegf*, *epo*, and *pdk1*, under 1% O_2_ conditions in ARPE19 cells (**B**). Note that gene expression of *hif-2a* was suppressed by fish ingredients. *Vegf*, *epo*, and *pdk1* were upregulated under 1% O_2_ conditions and significantly suppressed by fish ingredients administration. Fish ingredients were added at 1 mg/mL and the hypoxic conditions were maintained for 12 h. *n* = 6/group. * *p* < 0.05, ** *p* < 0.01, *** *p* < 0.001, † *p* < 0.0001, ‡ *p* < 0.00001 compared with 1% O_2_/vehicle. Error bars indicate mean plus SD. Veh., vehicle; Topo, topotecan; epo, erythropoietin.

**Figure 3 nutrients-12-01055-f003:**
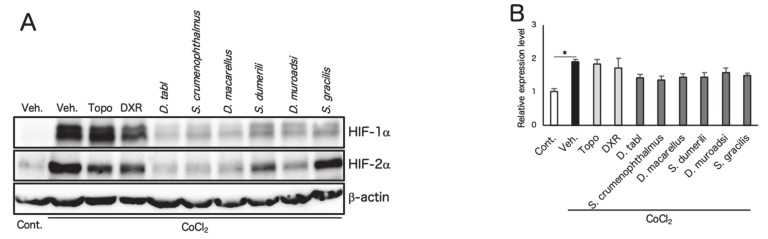
Fish ingredients suppress HIF-1a protein expression in vitro. Western blotting for HIF-1α and HIF-2α was performed under CoCl_2_ condition in ARPE19 cells (**A**), and for HIF-1α under 1% O_2_ condition in ARPE19 cells (**D**), under CoCl2 (**F**) or 1% O2 conditions (**H**) in 661W cells. Quantification of the blots showed that the administration of fish ingredients suppressed increased HIF-1α protein expression under CoCl_2_ (**B**) or 1% O_2_ conditions (**E**) in ARPE19 cells and under CoCl_2_ (**G**) or 1% O_2_ conditions (I) in 661W cells (*n* = 3). Quantification of the blots also showed that the administration of fish ingredients suppressed the increased HIF-2α protein expression under CoCl_2_ in ARPE19 cells (**C**) (*n* = 3). CoCl_2_ was administered at a concentration of 200 mM, fish ingredients were added at 1 mg/mL simultaneously, and cells were incubated for 24 h. The hypoxic conditions were maintained for 48 h. Note that the fish ingredients inhibited HIF-1α and HIF-2α expression induced by CoCl_2_ or hypoxia. * *p* < 0.05, ** *p* < 0.01, *** *p* < 0.001 compared with 1% O_2_/vehicle or CoCl_2_/vehicle. Error bars indicate mean plus SD. Veh., vehicle; Topo, topotecan; DXR, Doxorubicin.

**Figure 4 nutrients-12-01055-f004:**
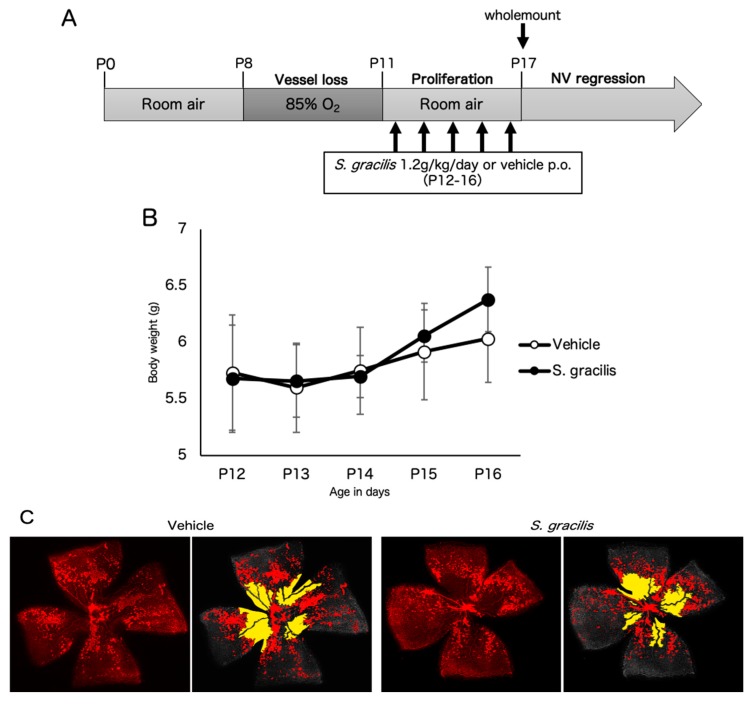
Oral administration of *Spratelloides gracilis* ingredients show a tendency to suppress retinal neovascularization in a murine oxygen-induced retinopathy (OIR) model. (**A**) A schematic illustration of the oxygen-induced retinopathy (OIR) procedure. Vehicle (*n* = 6) or *S. gracilis* (*n* = 5) was orally administered from P12 to P16. (**B**) Mean body weight change of mice. The average body weight of the *S. gracilis* group did not differ from that of the vehicle group through the administration period. (**C**) Representative images of retinal vasculature staining with isolectin B4. Areas of neovascular tufts (red) or vaso-obliteration (yellow) are highlighted. (**D**) Quantification of neovascular tufts and vaso-obliteration (right) in OIR as a percentage of the total retinal area. Note that *S. gracilis* tended to suppress neovascular tufts while no significant difference was observed in vaso-obliteration. n.s., not significant. Error bars indicate mean plus SD.

**Figure 5 nutrients-12-01055-f005:**
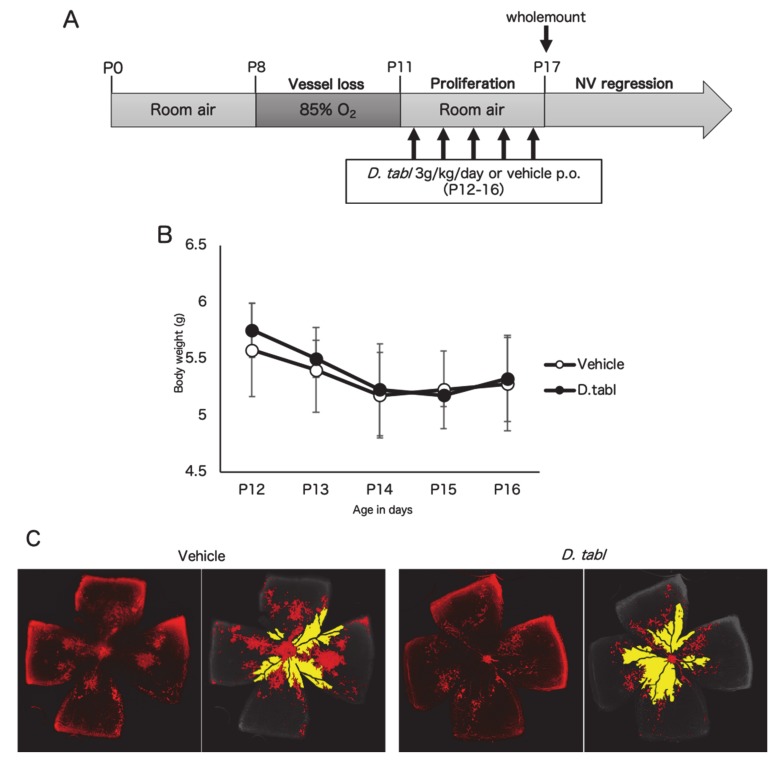
Oral administration of *Decapterus tabl* ingredient suppress retinal neovascularization in a murine oxygen-induced retinopathy (OIR) model. (**A**) Schematic illustration of the OIR procedure. Vehicle or *D. tabl* was orally administered from P12 to P16. (**B**) Mean body weight change of mice. The average body weight of the *D. tabl* group did not differ from that of the vehicle group throughout the administration period. (**C**) Representative images of retinal vasculature staining with isolectin B4. Areas of neovascular tufts (red) or vaso-obliteration (yellow) are highlighted. (**D**) Quantification of neovascular tufts (left) and vaso-obliteration (right) in OIR as a percent of the total retinal areas. Note that *D. tabl* significantly (*p* < 0.05) suppressed neovascular tufts, while no significant difference was observed in vaso-obliteration. *n* = 4/group. * *p* < 0.05; n.s., not significant. Error bars indicate mean plus SD.

**Table 1 nutrients-12-01055-t001:** The list of fishes showing hypoxia-inducible factor (HIF) inhibitory effects in the second screening with statistical analysis and the rate of change of HIF activity compared with dimethyloxalylglycine (DMOG)-administrated controls (*n* = 3). (^†^ Positive control chemicals) *** *p* < 0.001, † *p* < 0.0001, ‡ *p* < 0.00001 compared with DMOG.

Species of Fish	Japanese Name	Sampling Parts	Rate of Change (%)	*p* value
Topotecan ^†^	-	-	−54.4	0.0000000 ‡
Doxorubicin ^†^	-	-	−54.4	0.0000000 ‡
*Selar crumenophthalmus*	Meaji	skinless fillet	−47.2	0.0000001 ‡
*Seriola dumerili*	Kanpachi	white muscle, dorsal	−29.0	0.00003 †
*Spratelloides gracilis*	Kibinago	headless	−27.9	0.0003 ***
*Decapterus macarellus*	Kusayamoro	skinless fillet	−24.4	0.0001 ***
*Panulirus japonicus*	Ise-ebi	muscle, abdomen	−17.2	0.069
*Sulculus diversicolo supertexta*	Tokobushi	Muscle, foot	−12.2	0.283
*Trachurus japonicus*	Maaji	skinless fillet	−8.8	0.127
Dried bonito	Ara-bushi	-	−3.8	0.999
*Scomberoides lysan*	Ikekatsuo	skinless fillet	−2.9	0.999
*Rhabdosargus sarba*	Hedai	white muscle, dorsal	−1.4	0.999
DMOG	-	-	0	-

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
