# Peer review of "Hypoxia-Inducible Factor Inhibitors Derived from Marine Products Suppress a Murine Model of Neovascular Retinopathy"

_nutrients, 2020, doi:10.3390/nu12041055_

Round 1
Reviewer 1 Report
General comments
Hypoxia-inducible factor inhibitors derived from marine products suppress a murine model of Neovascular Retinopathy by Shoda et al. submitted for publication in Nutriens is an interesting manuscript with novel findings.
The authors have used versatile cell biological methods in vitro completed with an animal model. They have explored water-soluble-ingredients from 68 marine fish species showing HIF inhibitory effects in cell culture and additionally, and when administered the strongest candidate in a form of pellet into experimental animals they were able to show this particulate extract to suppress retinal neovascular tufts.
The scientific approach appears to be almost identical to that published previously in Int. J. Mol. Sci. (IF 4,183) by the same group (Kunimi et al. 2019); in this study they used an original library of natural compounds consisting of 238 plant and food extracts.
My main concern deals with the extracts they used. In figures and tables the authors call them either “sampling part” or “fish ingredients” giving no description whatsoever about the composition of the extracts. In Discussion, they consider these extracts to contain small molecules such as dipeptides, amino acids, nucleic acids or minerals. In addition, dose-response curves are not shown for any of the extracts. The exctraction methods may have excluded a number of biologically active substances.
It is unclear why the authors decided to use that particular dose when they administered the extracts into animals and no information is given as to how the physiology of animals responded to these extracts in general.
Please discuss these issues in a more detailed manner.
Reviewer 2 Report
The study by Shoba et al. evaluated the anti-angiogenic effects of HIF inhibitors derived from marine products in a mouse model of OIR. The study is well-performed. Data are well-presented and add to the current literature. However, there are some minor concerns.
- It wasn't clear in the manuscript how were fish products administered orally. Was it given to their nursing mothers via drinking mother? If so, do authors know the exact amounts of fish products the pups received?
- VEGF is the primary factor driving the formation of neovascular tufts in the OIR model. Can the authors measure VEGF levels in the retina of OIR mice treated with the fish products? New data on VEGF will further support their anti-angiogenic effects.
